# Pharmacological inhibition of CLK2 activates YAP by promoting alternative splicing of AMOTL2

**Maya L Bulos[1], Edyta M Grzelak[1], Chloris Li-Ma[1], Emily Chen[2], Mitchell Hull[2], Kristen A Johnson[2], Michael J Bollong[1]***

[1]Department of Chemistry, The Scripps Research Institute, La Jolla, United States; [2]Calibr, A Division of Scripps Research, La Jolla, United States

**Abstract** Yes-associated protein (YAP), the downstream effector of the evolutionarily conserved Hippo pathway, promotes cellular proliferation and coordinates certain regenerative responses in mammals. Small molecule activators of YAP may, therefore, display therapeutic utility in treating disease states involving insufficient proliferative repair. From a high-throughput chemical screen of the comprehensive drug repurposing library ReFRAME, here we report the identification of SM04690, a clinical stage inhibitor of CLK2, as a potent activator of YAP-driven transcriptional activity in cells. CLK2 inhibition promotes alternative splicing of the Hippo pathway protein AMOTL2, producing an exon-skipped gene product that can no longer associate with membrane-bound proteins, resulting in decreased phosphorylation and membrane localization of YAP. This study reveals a novel mechanism by which pharmacological perturbation of alternative splicing inactivates the Hippo pathway and promotes YAP-dependent cellular growth.

***For correspondence:**
mbollong@scripps.edu

**Competing interest:** The authors declare that no competing interests exist.

## eLife assessment

This paper reports **important** findings on a potent activator of the YAP pathway, demonstrating its mechanism through alternative splicing changes. The authors provide **convincing** evidence to support their claims. This research is of interest to biologists studying alternative splicing or the Hippo pathway, with significant implications for medical research.

## Introduction

The evolutionarily and functionally conserved Hippo signaling pathway functions as a master regulator of cellular proliferation and tissue growth in animals. Molecularly, the Hippo pathway is composed of a kinase cascade that catalyzes a series of phosphorylation events at the plasma membrane. The core pathway includes Ste20-like kinase 1 (MST1) and MST2, which phosphorylate and activate large tumor suppressor kinase 1 (LATS1) and LATS2 (*Wu et al., 2003*; *Xu et al., 1995*). LATS1/2 phosphorylation results in the phosphorylation of downstream effector Yes-associated protein 1 (YAP), leading to its cytoplasmic retention and inactivation (*Huang et al., 2005*). Several other proteins work together to regulate the core kinase cascade, such as Salvador homolog 1 (SAV1) and MOB kinase activator 1 A and 1B (MOB1A/B), binders of MST1/2 and LATS1/2, respectively (*Udan et al., 2003*; *Lai et al., 2005*). In addition to phosphorylation-dependent regulation of the Hippo pathway, YAP is also regulated by protein-protein interactions, such as with the angiomotin family of proteins (AMOT, AMOTL1, and AMOTL2), which directly bind to and cause the localization of YAP at tight junctions (*Wang et al., 2011*). Adherens junctions and tight junctions are important sites of Hippo pathway regulation that link to the core kinases of the Hippo pathway through the AMOT proteins. When the Hippo pathway

is inactivated, YAP translocates to the nucleus, acting as a transcriptional co-activator of TEA domain transcription factors (TEADs), which increases the transcription of genes involved in cellular proliferation and survival.

Regulation of the Hippo pathway by cell-cell contacts, metabolic changes, and diverse extrinsic inputs results in decreased activity of YAP, which prevents organ overgrowth and abnormal cellular proliferation (*Yu and Guan, 2013*). On the other hand, YAP activity is also critical in healing injury by augmenting cellular proliferation and survival. While many multicellular organisms can repair wounded or damaged organs, mammalian regeneration is more limited, as many adult tissues lack the capacity to regenerate. Genetic activation of YAP, either by genetic inactivation of upstream Hippo pathway proteins or by forced overexpression of YAP, has been shown to enhance endogenous repair in wounded tissues. For example, hepatic knockdown of MST1/2 with small interfering RNA (siRNA) in the mouse after partial hepatectomy augments regeneration of the liver via YAP-dependent hepatocyte proliferation (*Loforese et al., 2017*). Likewise, in the skin, YAP is essential for regenerative wound repair, as siRNA-mediated knockdown of YAP markedly delays wound closure (*Lee et al., 2014*). Conversely, conditional hyperactivation of YAP enhances the proliferation of epidermal keratinocytes, thickening the epidermal layer of neonatal mice (*Zhang et al., 2011*). Lastly, in the heart, an organ previously considered to have no regenerative potential post-development, conditional SAV deletion promotes the proliferation of cardiomyocytes and reverses deficits in ejection fraction volume post-myocardial infarction (*Leach et al., 2017*). Collectively, these studies suggest that small molecule activators of YAP may, therefore, display therapeutic utility in treating diseases etiologically defined by an insufficient proliferative response.

We recently reported the identification of PY-60, a small molecule that activates YAP by binding to a membrane-binding scaffolding protein, annexin A2 (ANXA2), which we defined as a central component of the Hippo pathway (*Shalhout et al., 2021*). This study suggested to us that likely many other druggable mechanisms exist for activating YAP including those associated with previously drugged cellular proteins. Here, to explore the hypothesis that known drugs might activate YAP-driven transcriptional activity, we performed a reporter-based screen of the comprehensive drug repurposing library ReFRAME. We found that SM04690, a small molecule kinase inhibitor in clinical trials for the treatment of osteoarthritis, activates YAP with nanomolar potency in cells. Mechanistic studies revealed the relevant target kinase of SM04690 to be CDC-like Kinase 2 (CLK2), inhibition of which induces alternative splicing of Hippo signaling component AMOTL2 that can no longer localize YAP to the plasma membrane, in effect activating YAP.

## Results
### Screening of the ReFRAME library

To identify small molecule activators of YAP, we screened the Repurposing, Focused Rescue, and Accelerated Medchem (ReFRAME) small molecule library using a HEK293A cell line with a stably integrated cassette of eight copies of the TEAD-binding element upstream of luciferase (293A-TEAD-LUC). This library contains ~13,000 small molecules that have undergone substantial preclinical evaluation and clinical development are FDA-approved (*Figure 1A*; *Janes et al., 2018*). The screen was conducted using a previously validated assay in 1,536-well format at three compound concentrations (200 nM, 2 μM, and 10 μM) to maximize the potential number of mechanisms surveyed. Hit compounds were defined if they displayed increases in TEAD-LUC signal of more than five robust Z scores above the mean plate signal (*Figure 1B*). Identified hits could be grouped into several classes of inhibitors including known nonspecific modulators of transcription (e.g. bromodomain inhibitors, histone deacetylase inhibitors), ion channel modulators, and kinase inhibitors, among others (*Figure 1C*). Of these, we identified 15 kinase inhibitors that did not have any associated YAP augmenting activity and are not characterized inhibitors of MST1/2 and LATS1/2. Evaluating these 15 hits in dose response using the original screening assay with library material, we confirmed nine displayed reproducible YAP activation (*Figure 1D*). Of these, three were inhibitors of non-receptor tyrosine kinases including Cerdulatinib (inhibitor of SYK and JAK paralogs *Coffey et al., 2014*; TEAD-LUC $EC_{50}$=0.31 μM), Peficitinib (a pan JAK inhibitor *Nakada and Oda, 2015*; TEAD-LUC $EC_{50}$=2.4 μM), and KX02 (an inhibitor of SRC family kinases *Tu et al., 2012*; TEAD-LUC $EC_{50}$=3.7 μM). Also identified as efficacious YAP activators were XL-999 (a pan RTK inhibitor, TEAD-LUC $EC_{50}$=1.5 μM) and CEP-1347 (a JNK family

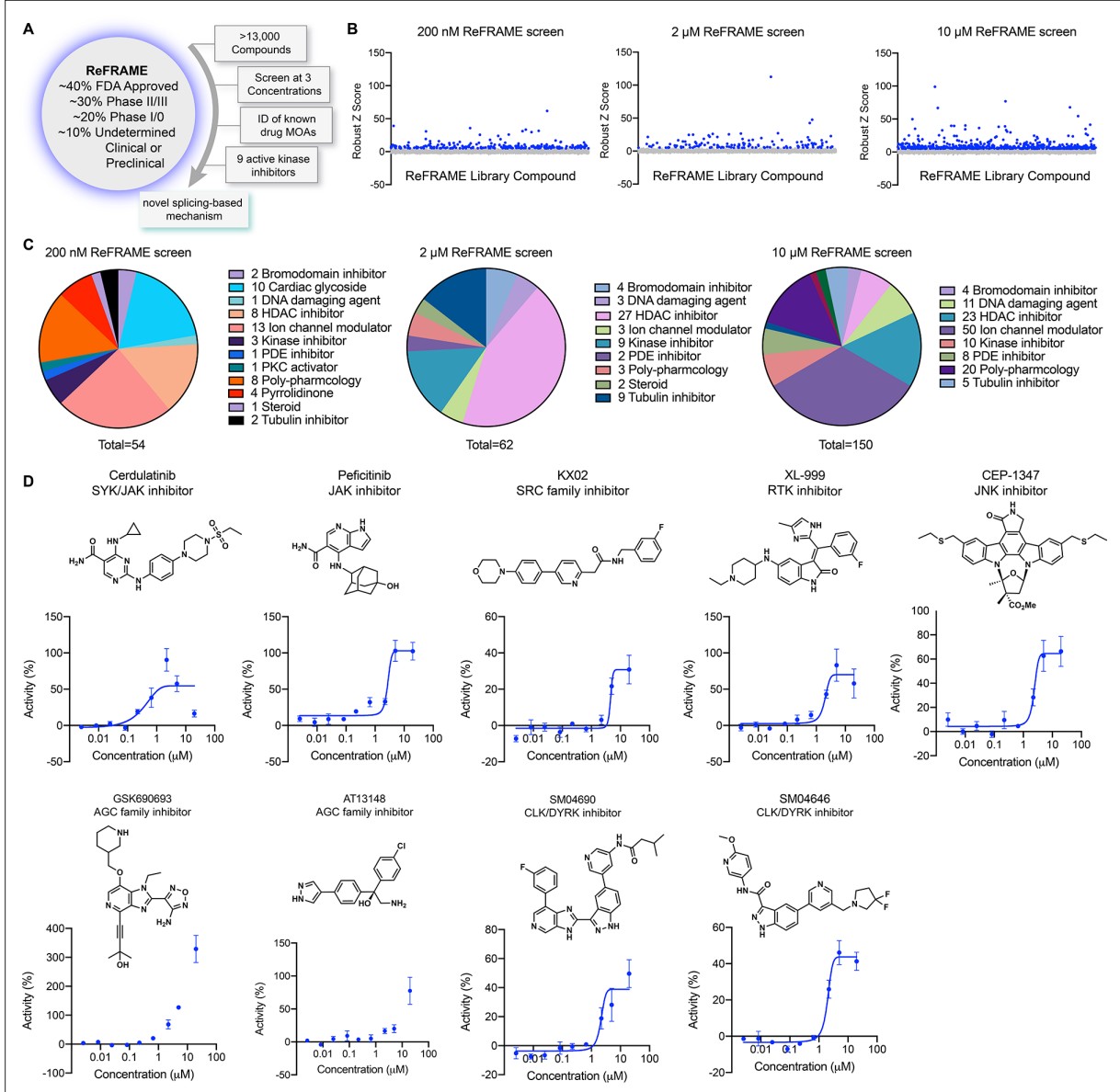

**Figure 1.** A screen of Repurposing, Focused Rescue, and Accelerated Medchem (ReFRAME) identifies pharmacological mechanisms for activating yes-associated protein (YAP). (**A**) Schematic depicting screening workflow, which resulted in the identification of nine kinase inhibitor-based YAP activating small molecules. (**B**) Scatter plots of TEAD-LUC activity from each of the screens were conducted at the final compound concentrations of 200 nM, 2 μM, and 10 μM. Each data point represents a single compound. Hits were classified as having robust Z scores over five. (**C**) Pie charts indicate the classes of small molecules characterized as hits on each screen. (**D**) Structure and dose-response plots of TEAD-LUC activity of the nine identified kinase inhibitors (n=3, mean and s.d.).

The online version of this article includes the following source data for figure 1:

**Source data 1.** Data for each Repurposing, Focused Rescue, and Accelerated Medchem (ReFRAME) screen in *Figure 2B*.

**Source data 2.** Dose response data for *Figure 2D*.

inhibitor; TEAD-LUC EC$_{50}$=2.5 μM). Two inhibitors of the AGC family of kinases were also identified, namely GSK690693 (*Rhodes et al., 2008*) (TEAD-LUC EC$_{50}$ >20 μM) and AT13148 (*Yap et al., 2012*) (TEAD-LUC EC$_{50}$ >20 μM). Lastly, we identified two inhibitors of CLK and DYRK family kinases including SM04690 (*Deshmukh et al., 2018*) (TEAD-LUC EC$_{50}$=5.7 μM) and SM04646 (TEAD-LUC EC$_{50}$=2.1 μM), which had been initially described as agents which inhibit canonical Wnt signaling and induce differentiation of chondrocytes. Among these, we chose to further characterize SM04690, which is in clinical trials for the treatment of osteoarthritis of the knee, as no association of its target

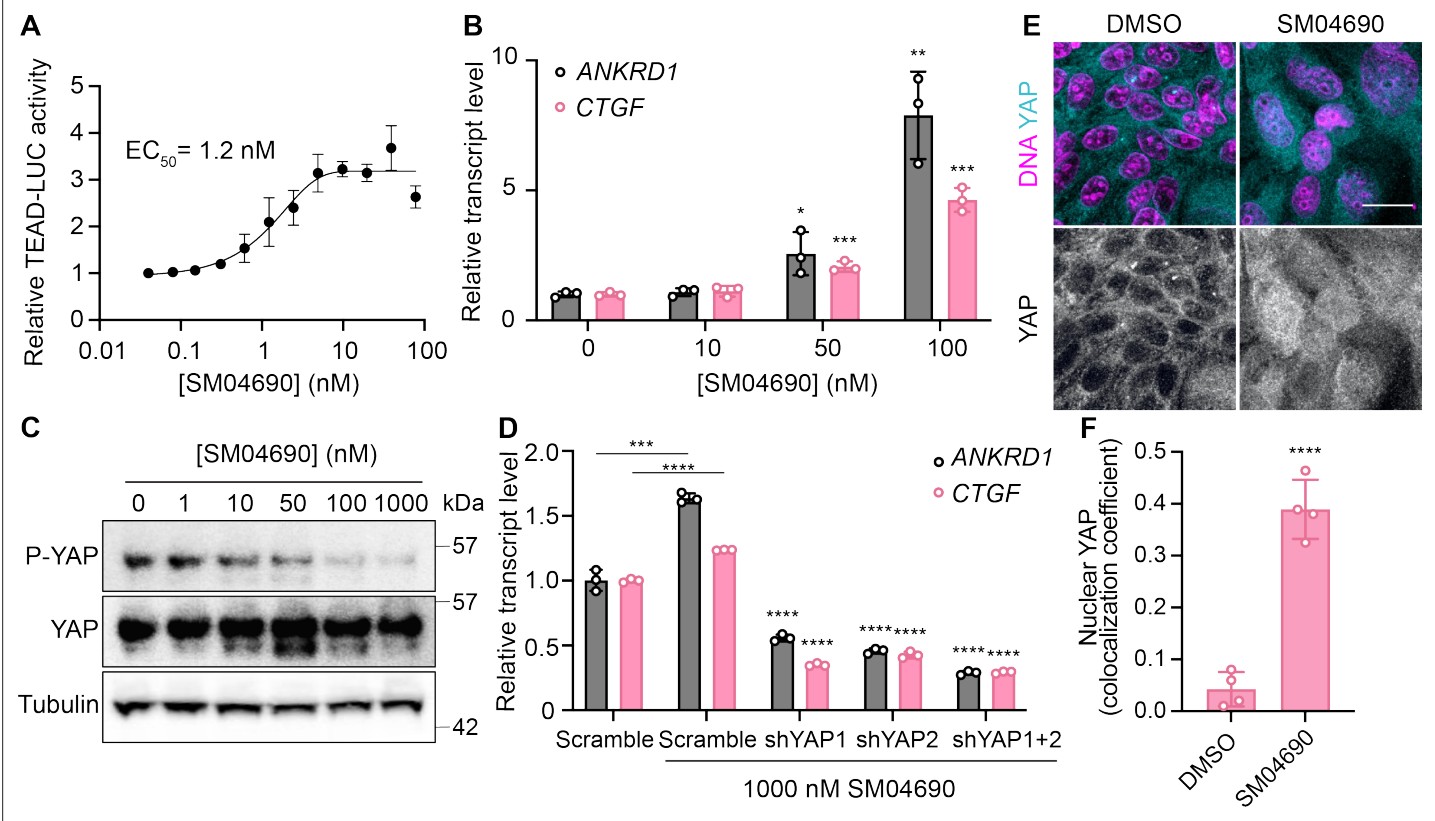

**Figure 2.** SM04690 is a pharmacological activator of yes-associated protein (YAP). (**A**) Relative luminance values after 24 hr treatment of 293A-TEAD-LUC cells with the indicated concentrations of SM04690 (n=3 biological replicates, mean and s.e.m.). (**B**) Relative transcript levels of YAP-dependent genes (*ANKRD1* and *CTGF*) from HEK293A cells were treated for 24 hr with SM04690, measured by RT-qPCR (n=3, mean and s.d.). (**C**) Western blot analysis of anti-phospho-YAP and total YAP levels from HEK293A cells treated with the indicated concentrations of SM04690 for 24 hr. (**D**) Relative transcript levels of *ANKRD1* and *CTGF* from HEK293A cells expressing the indicated YAP targeting shRNAs and then treated with 1 μM SM04690, measured by RT-qPCR (n=3, mean and s.d.). Asterisks above samples treated with shRNAs to YAP and SM04690 refer to statistical significance as compared to scramble treated with SM04690. (**E**) Representative images of anti-YAP immunofluorescent staining of MDCK cells following treatment with 1 μM SM04690. Anti-YAP antibody in teal and Hoechst 33342 in pink to visualize nuclei (scale bar = 10 μm). (**F**) Quantification of anti-YAP and Hoechst 33342 (DNA) correlative immunofluorescent staining (n=4, mean and s.d.). (**B, D, F**) Statistical analyses are univariate two-sided t-tests (*p<0.0332, **p<0.0021, ***p<0.0002, ****p<0.0001).

The online version of this article includes the following source data and figure supplement(s) for figure 2:

**Source data 1.** Data related to *Figure 2A, B and D*.

**Source data 2.** Uncropped blots for *Figure 2C*.

**Figure supplement 1.** SM04690 induces yes-associated protein (YAP)-dependent proliferation at non-toxic concentrations.

**Figure supplement 1—source data 1.** Relative luminance values for *Figure 2—figure supplement 1C*.

kinases had previously been reported as related to Hippo pathway signaling (*Deshmukh et al., 2018*; *Yazici et al., 2017*).

## SM04690 is a YAP activator

We first confirmed the YAP-stimulating activity of SM04690 in 384-well format using 293A-TEAD-LUC cells with fresh, commercially obtained material, finding that the compound dose-dependently increased TEAD-LUC reporter activity with an EC$_{50}$ of 1.2 nM, a value considerably more potent than that observed in the screening assay (*Figure 2A*). We next confirmed that SM04690 functionally activated a YAP driven transcriptional program, as SM04690 was found to significantly increase the levels of YAP-controlled transcripts *ANKRD1* and *CTGF* (*Figure 2B*). Likewise, SM04690 treatment markedly decreased phosphorylation of YAP, as SM04690 was found to decrease immunopositivity dose-dependently to anti-phospho-YAP in Western blotting experiments following

24 hr compound treatment (*Figure 2C*). We next knocked down YAP transcript using shRNA and found that 1 μM compound treatment failed to increase *ANKRD1* and *CTGF* transcript levels in the absence of YAP, suggesting the compound activates a YAP-dependent transcriptional program (*Figure 2D*). We additionally observed by immunofluorescent imaging that YAP localized to the nucleus in MDCK cells treated with SM04690 (1 μM, 24 hr), suggesting that compound treatment promotes nuclear translocation and activation of YAP. This is in comparison to diffuse plasma membrane and cytoplasm YAP localization in DMSO-treated cells (*Figure 2E and F*). Lastly, since pharmacological YAP activators have been shown to promote the expansion of cells, we tested 0.2 nM SM04690 treatment in immortalized human keratinocytes (HaCaT) over 7 days and found treatment promoted clonal expansion, as seen with rhodamine B staining and increased nuclei counts (*Figure 2—figure supplement 1A, B*). Importantly, these proliferative effects occur at a non-cytotoxic concentration (under 0.5 nM) of SM04690 treatment (*Figure 2—figure supplement 1C*).

## CLK2 is the relevant cellular target of SM04690

SM04690 was first described in the literature as an inhibitor of Wnt signaling capable of promoting differentiation of mesenchymal stem cells to chondrocytes (*Deshmukh et al., 2018*). Following work defined SM04690 as an inhibitor of several subfamilies of related CMGC family protein kinases (*Deshmukh et al., 2019*), including Dual Specificity Tyrosine Phosphorylation Regulated Kinase (DYRK) and CDC Like Kinase (CLK) subfamilies (*Figure 3—figure supplement 1A*). Specifically, SM04690 has been shown to inhibit CLK2, CLK3, DYRK1A and B, Homeodomain Interacting Protein Kinase 1 (HIPK1), and HIPK2 with over 90% inhibition at 500 nM (*Deshmukh et al., 2019*). As such, we hypothesized that inhibiting one of these targets was likely responsible for modulating YAP activity. Previously published transcript expression data from our lab indicated that among the targets of SM04690 only CLK2, DYRK1B, HIPK1, and HIPK2 are expressed in HEK293A cells, helping to narrow our search (*Figure 3—figure supplement 1B*; *Shalhout et al., 2021*). We then treated 293A-TEAD-LUC cells with structurally distinct small molecule inhibitors of these putative targets to see if other treatments of cells with inhibitors to the potential target might recapitulate the YAP activating effect of SM04690. We found that only inhibitors targeting CLK2 (CC671, CLK-in-T3, and T025) elicited a dose-dependent increase in YAP activity (*Figure 3—figure supplement 1C*). In contrast, inhibitors targeting DYRKs (Harmine, INDY, and Leucettine L41) and CLK4 (ML167, TG003) showed no activity at concentrations over multiple orders of magnitude (*Figure 3—figure supplement 1D*). To confirm CLK2 as the relevant target of SM04690, we knocked down CLK2 by approximately 50% using three different targeting shRNAs, all of which increased YAP-controlled transcripts *ANKRD1*, *CTGF*, and *CYR61* to varying degrees (*Figure 3A and B*). Collectively, these data strongly suggested that CLK2 was most likely the target of SM04690 responsible for activating YAP.

CLK2 is a protein kinase that phosphorylates serine/arginine-rich (SR) proteins of the spliceosomal complex, which play an important role in exon selection by the spliceosome (*Jeong, 2017*; *Long and Caceres, 2009*). The phosphorylation states of SR proteins dictate their subcellular localization and interactions with pre-mRNA (*da Silva et al., 2015*). When CLK2 is inhibited, SR proteins are not phosphorylated, leading to alternative splicing that is most often exon skipping (*Salvador and Gomis, 2018*). Previous work using pharmacological CLK2 inhibitors revealed that inhibiting SR phosphorylation had a marked transcriptome-wide effect on altering alternative splicing. Interestingly, the third most skipped exon of the transcriptome in response to CLK2 inhibition corresponded to exon 5 of Angiomotin-like 2 (AMOTL), as nearly 3,000 new junction reads between non-neighboring exons were observed in this study (*Araki et al., 2015*; *Figure 3—figure supplement 2*). Exon 9 of AMOTL2 was also skipped but to a lesser extent with around 700 new junction reads. AMOTL2 is part of the AMOT family of proteins and is crucial to the tight junction localization of YAP through direct interaction with YAP's WW domain (*Wang et al., 2011*). Notably, the knockdown of AMOTL2 decreases the content of tight junction-localized YAP, instead promoting YAP translocation to the nucleus (*Zhao et al., 2011*). Given the central role that AMOTL2 plays in regulating YAP as well as the observation that no other canonical Hippo pathway member displayed alternative splicing in response to CLK2 inhibition, we hypothesized that SM04690 most likely increased YAP activity through modulating AMOTL2 splicing.

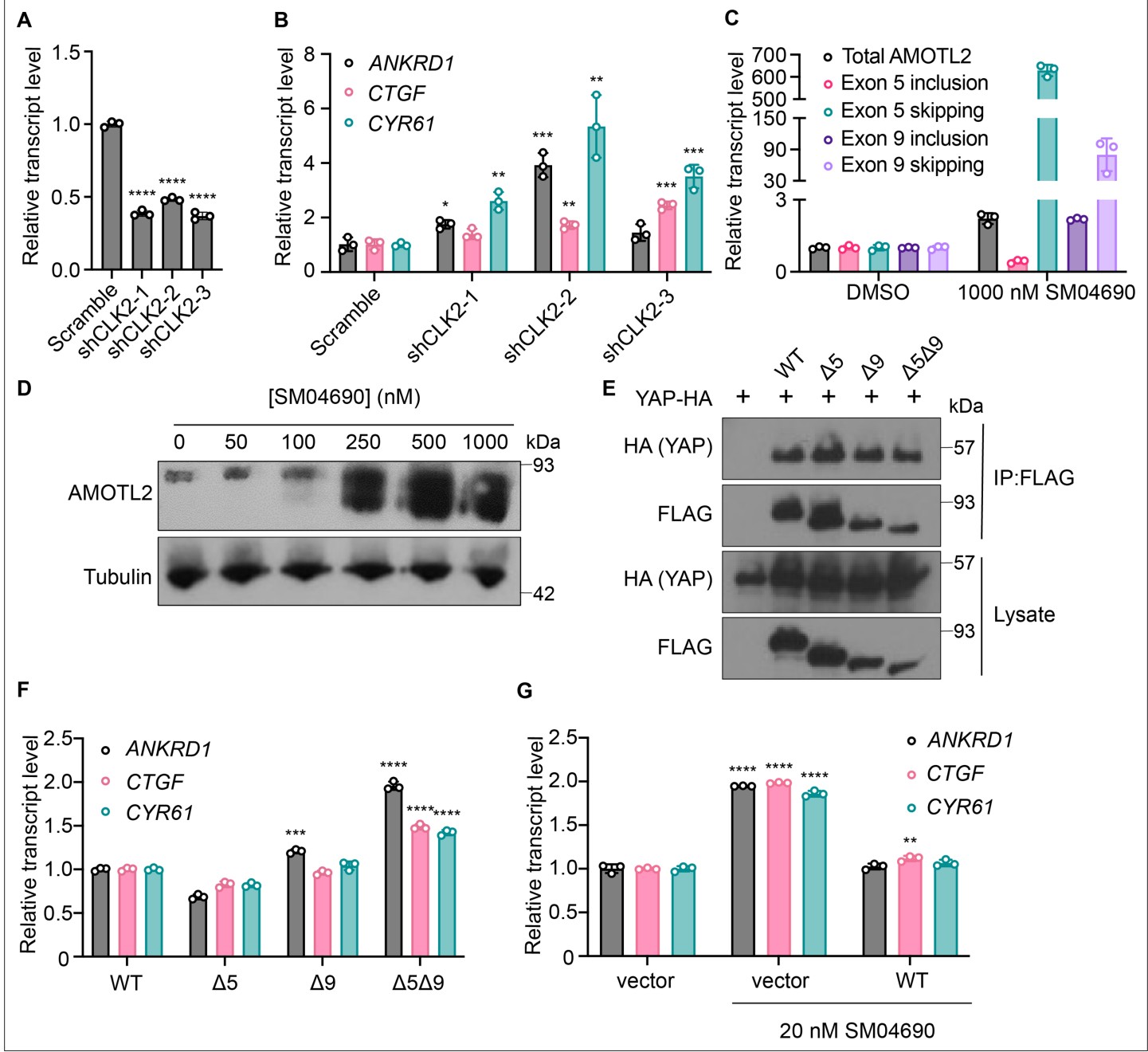

**Figure 3.** Inhibition of CLK2 by SM04690 induces alternative splicing of Hippo pathway component Angiomotin-like 2 (AMOTL2). (**A**) Relative transcript levels of CLK2 following knockdown by the indicated shRNA, measured by RT-qPCR (n=3, mean and s.d.). (**B**) Relative transcript levels of yes-associated protein (YAP)-dependent genes (*ANKRD1, CTGF, CYR61*) expressing the indicated shRNA, measured by RT-qPCR (n=3, mean and s.d.). (**C**) Relative transcript levels of exon-included and exon-skipped spliceoforms following treatment with SM04690 for 24 hr in HEK293A cells, measured by RT-qPCR (n=3, mean and s.d.). (**D**) Anti-AMOTL2 immunoblotting from HEK293A cells treated with SM04690 for 24 hr. (**E**) Immunoblotting analysis of HA-YAP from anti-FLAG immunoprecipitated content from HEK293T cells. (**F**) Relative transcript levels of YAP-dependent genes (*ANKRD1, CTGF, CYR61*) following overexpression of 1 μg of AMOTL2 WT, Δ5, Δ9, Δ5Δ9, measured by RT-qPCR (n=3, mean and s.d.). (**G**) Relative transcript levels of YAP-dependent genes (*ANKRD1, CTGF, CYR61*) following overexpression of 2 μg WT AMOTL2 and treatment with 20 nM SM04690 for 24 hr (n=3, mean and s.d.). (**A, B, F, G**) Statistical analyses are univariate two-sided t-tests (*p<0.0332, **p<0.0021, ***p<0.0002, ****p<0.0001).

The online version of this article includes the following source data and figure supplement(s) for figure 3:

**Source data 1.** Relative transcript values for *Figure 3A, B, C, F and G*.

**Source data 2.** Uncropped blots for *Figure 3D*.

*Figure 3 continued on next page*

*Figure 3 continued*

**Source data 3.** Uncropped blots for *Figure 3E*.

**Figure supplement 1.** CLK2 inhibition activates yes-associated protein (YAP).

**Figure supplement 1—source data 1.** Relative luminance values for *Figure 3—figure supplement 1C and D*.

**Figure supplement 2.** Top 40 transcripts with skipped exons in response to CLK2 inhibition.

**Figure supplement 3.** Inhibition of CLK2 by SM04690 causes alternative splicing of Angiomotin-like 2 (AMOTL2) exons 5 and 9.

**Figure supplement 3—source data 1.** Relative transcript values for *Figure 3—figure supplement 3A*.

**Figure supplement 3—source data 2.** Uncropped gel for *Figure 3—figure supplement 3D*.

## SM04690 induces alternative splicing of AMOTL2

We first confirmed that SM04690 treatment results in alternative splicing of AMOTL2. Consistent with reported literature, quantitative polymerase chain reaction (RT-qPCR) revealed that 24 hr SM04690 treatment at 1 µM induces exon skipping for exon 5 by 629-fold and exon 9 by 80-fold when compared to untreated samples (*Figure 3C*, *Figure 3—figure supplement 3B*). The skipping of exons 5 and 9 was observed as low as 10 nM with 24 hr treatment of SM04690 (*Figure 3—figure supplement 3A*). Using semi-quantitative reverse transcription PCR, in which primers are designed directly flanking an exon, we could visualize the presence of each splice isoform using gel electrophoresis. Exon 5, which is 93 base pairs long, has a PCR amplicon length of 132 base pairs when primer sequences are added on both ends (39 base pairs for both) (*Figure 3—figure supplement 3C*). Indeed, in the untreated samples, the amplicon ran through the gel to just over 100 base pairs. However, in the sample treated with 100 nM of SM04690, there are two PCR amplicons, one which ran the same distance as the untreated sample and a second band that ran well under 100 base pairs, as only the sequences of the primers were amplified, indicating skipping of exon 5 (*Figure 3—figure supplement 3D*). Exon 9 is 180 base pairs long with a PCR amplicon length of 218 base pairs with the primer sequences, (*Figure 3—figure supplement 3C*) which we see in the untreated sample. Like exon 5, in the 100 nM SM04690 treated sample, a second PCR amplicon unique to the treated sample ran under 100 base pairs, indicating the skipping of exon 9 (*Figure 3—figure supplement 3D*). This experiment thus confirmed that SM04690 treatment of HEK293A cells created exon skipping of exons as indicated by shorter PCR amplicons in treated samples differing from the untreated samples by exactly the length of the missing exon. The presence of exon skipped spliceoforms at the protein level was confirmed by anti-AMOTL2 Western blot using an antibody that can detect each possible spliceoform of AMOTL2. We found that with increasing SM04690 treatment in HEK293A cells for 24 hr, there was the appearance of broad lower molecular weight bands, consistent with the buildup of multiple forms of exon skipped AMOTL2 (*Figure 3D*). It is important to note that both the RT-qPCR and Western blot not only indicate the increase in alternative splicing but also an increase in the total AMOTL2 present with SM04690 treatment, as the relative transcript level of total AMOTL2 in SM04690 treated cells increased 2.3-fold in the RT-qPCR analysis and the Western blot showed increase bandwidth.

## Exon-skipped AMOTL2 activates YAP through a dominant mechanism

AMOTL2 interacts directly with YAP, helping to localize it to the plasma membrane. Accordingly, we next assessed if the exon-skipped gene products might interfere with the ability of AMOTL2 to bind YAP. Using FLAG-tagged transient overexpression constructs of wild-type AMOTL2 (WT), exon 5-skipped AMOTL2 (Δ5), exon 9-skipped AMOTL2 (Δ9), and double exon-skipped AMOTL2 (Δ5Δ9) (*Figure 3—figure supplement 3E*), we performed co-immunoprecipitation studies with HA-tagged YAP at high cell density in HEK293T cells. We found that the three AMOTL2 spliceoforms retained a similar ability to bind YAP as the WT AMOTL2 transgene (*Figure 3E*). As such, we next assessed if overexpression of the spliceoform transgenes themselves was sufficient to activate YAP. From experiments in HEK293A cells transiently overexpressing WT AMOLT2 or each of the three transgenes (AMOTL2 Δ5, AMOTL2 Δ9, AMOTL2 Δ5Δ9), AMOTL2 Δ5Δ9 uniquely displayed 1.5–2 times higher levels of YAP controlled transcripts in comparison to cell overexpressing WT AMOTL2 (*Figure 3F*). In contrast, no transcriptional induction was observed by expression of AMOTL2 Δ5 and only a minor induction of *ANKRD1* was observed for AMOTL2 Δ9, suggesting the AMOTL2 Δ5Δ9 spliceoform likely has a stronger YAP activating effect than either skipped exon alone. Given the expression of the

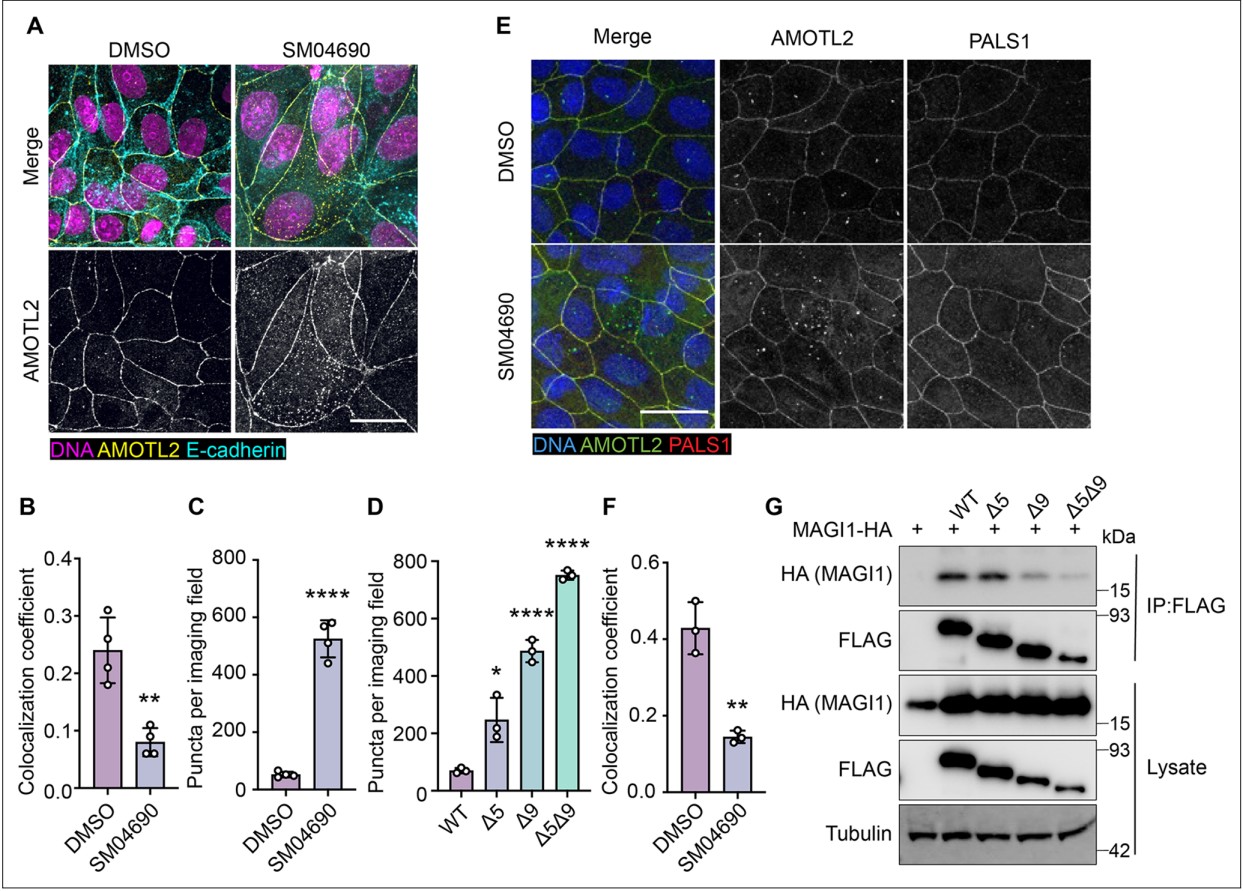

**Figure 4.** Alternative splicing of Angiomotin-like 2 (AMOTL2) promotes delocalization from the membrane, essential to the activity of SM04690. (**A**) Representative images of anti-AMOTL2 (yellow) and anti-E-cadherin (teal) immunofluorescent staining (with Hoechst 33342 in pink to visualize nuclei) of MDCK cells treated with control or 1 μM SM04690 for 24 hr (scale bar = 10 μm). (**B**) Quantification of anti-AMOTL2 and anti-E-cadherin correlative immunofluorescent staining (n=4, mean and s.d.). (**C**) Number of puncta in HEK293A cells treated with 1 μM SM04690 for 24 hr (n=4, mean and s.d.). (**D**) Number of puncta per image field in HEK293A cells from 12-well plates overexpressing AMOTL2 spliceoforms (n=3, mean and s.d.). (**E**) Representative images of anti-AMOTL2 (green) with anti-PALS1 (red) immunofluorescent staining of MDCK cells treated with control or 1 μM SM04690 for 24 hr. Hoechst 33342 (blue) was used to visualize nuclei (scale bar = 20 μm). (**F**) Quantification of anti-AMOTL2 and anti-PALS1 correlative immunofluorescent staining (n=3, mean and s.d.). (**I**) Immunoblotting analysis of HA-MAGI1 WW1, 2 from anti-FLAG immunoprecipitated content from HEK293T cells. (**B, C, D, F**) Statistical analyses are univariate two-sided t-tests (*p<0.0332, **p<0.0021, ****p<0.0001).

The online version of this article includes the following source data and figure supplement(s) for figure 4:

**Source data 1.** Uncropped blots for *Figure 4G*.

**Figure supplement 1.** Angiomotin-like 2 (AMOTL2) displays density-dependent localization to the plasma membrane.

**Figure supplement 2.** Angiomotin-like 2 (AMOTL2) spliceoforms are aggregation-prone and do not localize to membranes.

**Figure supplement 3.** SM04690 treatment delocalizes Angiomotin-like 2 (AMOTL2) from MAGI1.

AMOTL2 spliceoforms mimics the effects of SM04690 treatment, we sought to understand if overexpression of WT AMOTL2 might inhibit the ability of SM04690 to activate YAP. Indeed, overexpression of WT AMOTL2 was found to inhibit the upregulation of *ANKRD1, CTGF*, and *CYR61* in response to 20 nM compound treatment (*Figure 3G*), indicating that inducing alternative splicing is essential to the mechanism of SM04690.

## Alternative splicing of AMOTL2 promotes membrane delocalization

Consistent with previous research on the subcellular localization of AMOT family proteins (*Zhao et al., 2011*; *Wells et al., 2006*; *Mana-Capelli et al., 2014*), we found that AMOTL2 localizes to tight junctions at the plasma membrane of MDCK cells in conditions of high cell density (*Figure 4—figure supplement 1A, B*). Interestingly, we found that when MDCK cells were plated at high density and

then treated with 1 µM SM04690 for 24 hr, AMOTL2 was much less efficiently localized to the plasma membrane as assessed by immunofluorescent staining followed by confocal microscopy (*Figure 4A*). Indeed, SM04690-treated samples displayed a greater than twofold loss of colocalization of AMOTL2 with adherens junction protein E-cadherin (*Figure 4B*). Instead, SM04690 treatment resulted in the noticeable formation of AMOLT2 positive puncta in the cytoplasm of cells. This effect on AMOTL2 puncta formation could be additionally recapitulated in HEK293A cells (*Figure 4C*, *Figure 4—figure supplement 2A*), as 1 µM SM04690 resulted in a 10-fold increase in puncta. Likewise, we found that overexpression of AMOTL2 Δ5, AMOTL2 Δ9, and AMOTL2 Δ5Δ9 in HEK293A cells displayed dramatic increases in puncta number in comparison to WT AMOTL2 expression, with over 10-fold increases in puncta for AMOTL2 Δ5Δ9 compared to WT AMOTL2 (*Figure 4D*, *Figure 4—figure supplement 2B*).

We next sought to understand if the change of AMOTL2 localization to the cytoplasm might be responsible for the mechanism of YAP activation by SM04690. AMOTL2 was originally discovered as an interactor of the membrane-associated protein Membrane Associated Guanylate Kinase, WW, and PDZ Domain Containing 1 (MAGI1), and it has also been shown to interact with tight junction-associated proteins such as Protein Associated with LIN7, MAGUK p55 Family Member (PALS1) (*Patrie, 2005*; *Ernkvist et al., 2009*). Through these interactions, AMOTL2 localizes YAP to tight junctions, where YAP is inactivated by phosphorylation. We hypothesized that AMOTL2 spliceoforms created after SM04690 treatment can likely no longer interact with these membrane-associated proteins, acting to sequester YAP away from inhibitory Hippo signaling. Indeed, co-immunofluorescent studies in MDCK cells plated at high density indicated that 1 µM SM04690 treatment for 24 hr decreases co-localization of AMOTL2 with PALS1 and MAGI1 by approximately twofold (*Figure 4E and F*, *Figure 4—figure supplement 3A, B*). While cloning of these large membrane-associated proteins proved difficult, we were able to clone the WW domain regions of MAGI1, which are responsible for its interaction with AMOTL2. From co-immunoprecipitation with the FLAG-tagged AMOTL2 spliceoforms in HEK293T cells, we found that AMOTL2 Δ9 and AMOTL2 Δ5Δ9 lost their ability to interact with the WW domain regions of MAGI1 (*Figure 4G*). Collectively, these data suggest that alternative splicing of AMOTL2

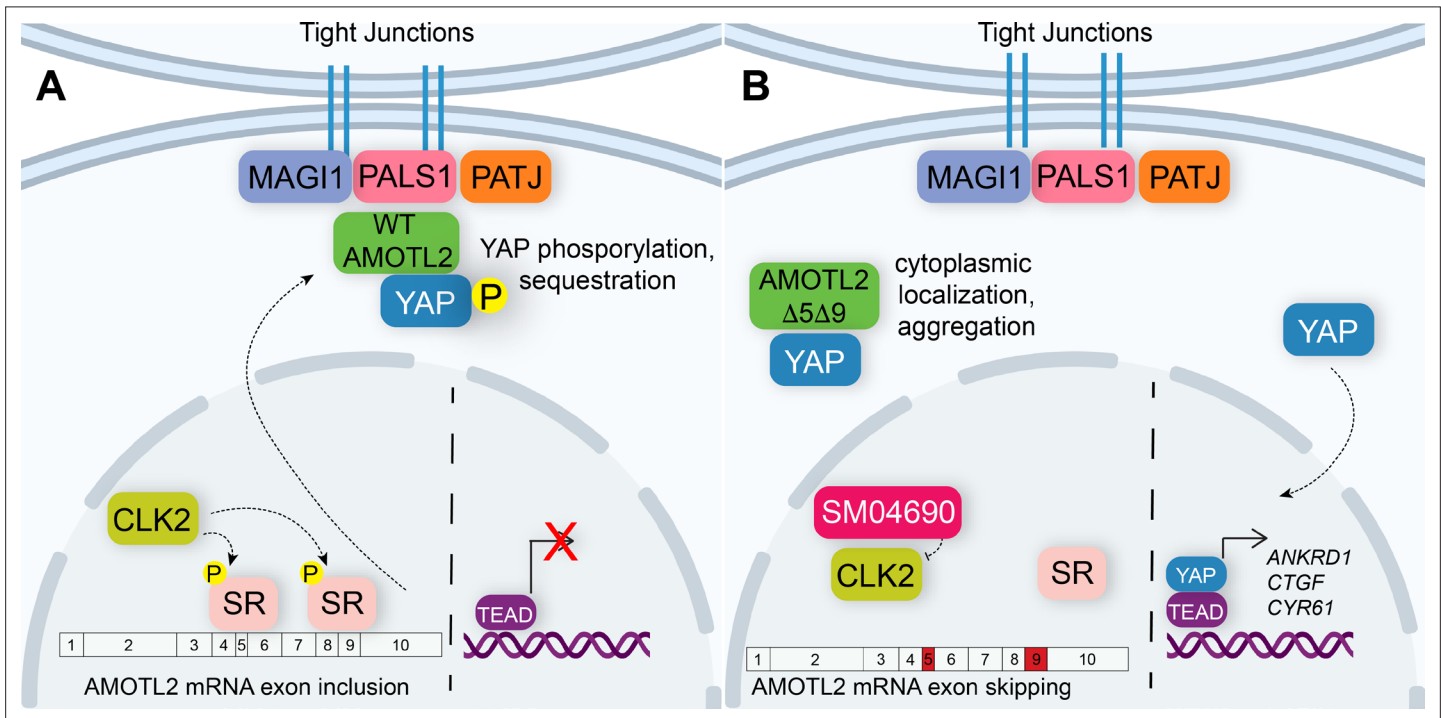

**Figure 5.** Alternative splicing of Angiomotin-like 2 (AMOTL2) modulates Hippo pathway activity. (**A**) A mechanistic model depicting that in the unperturbed system, AMOTL2 localizes at the membranes with proteins associated at tight junctions, bringing yes-associated protein (YAP) in contact with the core kinase cascade of the Hippo pathway resulting in its phosphorylation and retention in the cytoplasm. (**B**) Inhibition of CLK2 by SM04690 decreases phosphorylation of serine/arginine-rich splicing factors, leading to exon skipping of exons 5 and 9 of AMOTL2. AMOTL2 lacking exons 5 and 9 cannot interact with membrane and membrane-associated tight junction proteins. As such, AMOTL2 cannot localize YAP to tight junctions, and so YAP localizes to the nucleus, interacts with TEA domain transcription factors (TEADs), and activates its transcripts.

culminates in gene products less capable of plasma membrane localization, which results in decreased YAP phosphorylation.

## Discussion

In this study, we set out to identify repurposed drugs that might activate YAP, an effort we hypothesized would yield new druggable mechanisms for inactivating the Hippo pathway. From a high-throughput screen of the ReFRAME library, we have identified several kinase inhibitors that activate YAP. We specifically examined SM04690 as a novel YAP activator and have demonstrated a connection between CLK2 inhibition and YAP activity. Specifically, CLK2 inhibition induces exon skipping of exons 5 and 9 of AMOTL2, a protein integral to Hippo pathway activity. We propose a working model in which the alternative splicing of AMOTL2 impedes its ability to interact with proteins at the plasma membrane, such as PALS1 and MAGI-1 (*Figure 5*). The exon-skipped AMOTL2 gene product is, therefore, unable to shepherd YAP to tight junctions at the plasma membrane where it receives inhibitory phosphorylation. Consequently, unphosphorylated YAP is capable of translocating to the nucleus, where it interacts with TEADs to promote transcription of genes related to proliferation and survival. While we focused our efforts on understanding the role of SM04690, we identified several other kinase inhibitors as efficacious activators of YAP. Several of the reported targets of these identified inhibitors corresponded to related kinase families, including JAK kinases and AGC kinases. Future investigation into these and other protein kinase-based mechanisms will likely reveal several other druggable kinases that regulate Hippo pathway activity.

AMOTL2 serves a central role in regulating the Hippo pathway. Specifically, AMOTL2 has been shown to inhibit YAP, restraining it to the plasma membrane while concurrently stimulating the enzymatic activity of LATS1/2 (*Paramasivam et al., 2011*). These functions are possible due to AMOTL2's activity as a scaffolding protein, bringing LATS1/2 in close proximity to MST1/2 at the plasma membrane (*Mana-Capelli and McCollum, 2018*). While most research indicates AMOTL2 plays a role in negatively regulating YAP activity, there remains some debate in the field as to whether AMOTL2, and more broadly the AMOT family of proteins, might also exert YAP stimulating activities in the cell (*Lv et al., 2017*). The work presented here adds to the mounting evidence that AMOTL2 acts as a negative regulator of YAP, most prominently displayed through an attenuation of transcription of YAP-controlled genes when wild-type AMOTL2 is overexpressed in the context of the YAP-activating compound SM04690 (*Figure 3G*). A lack of scaffolding functionality by AMOTL2 spliceoforms lacking exons 5 and/or 9 may also help to explain how this alternative splicing is able to activate YAP. It has been shown that the AMOTL2-related protein AMOT regulates Hippo pathway activity through the formation of biomolecular condensates, in contrast to traditional scaffolding proteins (*Wang et al., 2022*). AMOT-laden condensates have been shown to concentrate core Hippo pathway kinases, acting to potentially sequester YAP in the cytoplasm. Since there is considerable redundancy between AMOT and AMOTL2 (*Mana-Capelli and McCollum, 2018*; *Li et al., 2012*; *Hirate and Sasaki, 2014*; *Leung and Zernicka-Goetz, 2013*), it is conceivable that AMOTL2 may also regulate YAP through biomolecular condensates. The exon skipped AMOTL2 spliceoforms identified here could have impaired ability to phase separate into condensates or fail to incorporate core Hippo pathway kinases into these condensates. Future studies examining the capacity of AMOTL2 to phase separate and interact with Hippo pathway proteins could reveal exactly how the lack of tight junction localization of the AMOTL2:YAP complex leads to increased YAP activity. More generally, this study highlights the existing need to precisely define the specific roles for each AMOT family member (i.e. AMOT, AMOTL1, and AMOTL2), delineating which functions are redundant and which are unique to each family member.

Others have also shown that alternative splicing influences YAP activity. Indeed, several core Hippo pathway proteins have alternatively spliced isoforms. For example, tissue-specific alternative splicing of YAP itself can lead to eight different isoforms and has been shown to regulate its activity by altering the efficiency with which it interacts with LATS1/2 and TEADs, leading to cell-type specific effects on cellular proliferation (*Gaffney et al., 2012*; *Iglesias-Bexiga et al., 2015*; *Vrbský et al., 2021*). In addition, LATS1 and LATS2 exist as 7 and 2 spliceoforms in the cell, respectively, affecting their interactions with Hippo pathway scaffolding proteins Neurofibromin 2 (NF2) and MOB1 and modulating their catalytic activity (*Porazinski and Ladomery, 2018*; *Kulaberoglu et al., 2017*). While the exon skipping observed in this study is derived from pharmacological perturbation, it is possible that similar processes exist endogenously, particularly in cell types or physiological scenarios with lower CLK2

activity. For example, several breast cancer cell lines have been found to display highly variable levels of CLK2 expression (*Yoshida et al., 2015*). As YAP hyperactivity has been characterized as essential in several human tumor types, including breast cancer (*Zanconato et al., 2016*), it follows that cancer cell lines with lower CLK2 activity may display increased aggressiveness by activating YAP through the splicing-based mechanism presented here.

One final consideration raised in this study concerns the use of alternative splicing modulators in the treatment of disease. SM04690 was initially reported in the literature as an inhibitor of canonical Wnt signaling, displaying an EC$_{50}$ of 19.5 nM in reporter-based assays (*Deshmukh et al., 2018*). Here, we have shown that SM04690 activates YAP with an EC$_{50}$ of 1.2 nM in TEAD-LUC assays, suggesting that several transcriptional programs including YAP might be modulated by this compound at therapeutic doses. Indeed, while SM04690 has been used in a localized fashion to treat osteoarthritis of the knee (*Yazici et al., 2020*; *Tambiah et al., 2021*), our work suggests that the use of broad-acting modulators of alternative splicing should be met with caution, as inhibition of the CLK family of kinases induces alternative splicing in many transcripts (*Araki et al., 2015*), which may lead to undesired off-target effects. Nevertheless, this work has identified that CLK2 inhibitors are novel pharmacological tools to study the processes governing the regulation of the Hippo pathway and it further underscores the utility of performing unbiased screens for identifying important cellular biological mechanisms.

## Materials and methods

### Cell culture

Reporter cells (293A-TEAD-LUC (HEK-293A-8xGTII-Luc)) were a gift from the laboratory of Xu Wu (Mass General). HEK293A cells were from Thermo Fisher, Waltham, MA. MDCK and HEK293T cells were obtained from the American Type Culture Collection (ATCC, Manassas, VA). HaCaT immortalized human keratinocytes were from AddexBio. All cell lines were cultured in DMEM (Corning, Corning, NY) supplemented with 10% FBS (Gibco, Grand Island, NY) and 1% penicillin/streptomycin (Gibco) and maintained at 37 °C with 5% CO$_2$.

### High-throughput screening

20 nL of stock compounds solvated in DMSO were dispensed using an Echo Acoustic Liquid Handler instrument (Labcyte, San Jose, CA) before adding 750 293A-TEAD-LUC cells per well in white 1536-well plates (Corning). 24 hr later, 2 μL of ONE-glo luciferase detection reagent (Promega, Madison, WI) was dispensed per well and the luminance signal was recorded with an Envision plate reader (Perkin Elmer, Waltham, MA).

### Miniaturized reporter assays

2500 293A-TEAD-LUC cells were plated in 50 μL of growth medium without FBS in white 384-well plates (Corning) and allowed to proliferate for 48 hr until fully confluent. 100 nL of compound solvated in DMSO was then transferred to each well using a Bravo Automated Liquid Handling Platform (Agilent, Santa Clara, CA) affixed with a pintool head (V&P Scientific, San Diego, CA). For TEAD-LUC activity assays, 30 μL of BrightGlo reagent solution (Promega, diluted 1:3 in water) was added to each well after 24 hr of compound treatment. For cytotoxicity assays, 30 μL of CellTiter-Glo reagent solution (Promega, diluted 1:6 in water) was added to each well after 24 hr and 72 hr of compound treatment. Luminance values were recorded on an Envision plate reader (Perkin Elmer).

### shRNA knockdown studies

CLK2-targeting shRNA vectors 1, 2, and 3 refer to Sigma Mission shRNA lentiviral clones TRCN0000197205, TRCN0000229970, TRCN0000355628, respectively. YAP-targeting shRNA vectors 1 and 2 refer to pLKO1-shYAP1 (Addgene, #27368) and pLKO1-shYAP2 (Addgene, #27369), respectively. The non-targeting scrambled control vector refers to SHC002 (Sigma, Burlington, MA). Lentiviruses were generated in HEK293T cells by transient expression of the vectors with pSPAX2 and pMD2.G packaging vectors (Addgene plasmids #12260 and #12259). Viral supernatants were collected after 48 hr of expression and passed through a 45 μm syringe filter before exposure to target cells.

## Quantitative reverse transcription PCR (qRT-PCR)

After 48 hr of transgene expression or 24 hr of compound treatment, cells were collected by trypsinization and subsequent centrifugation at 1200 × $g$. RNA was isolated using an RNeasy kit (Qiagen, Germantown, MD) and RNA concentrations were determined using a NanoDrop instrument. 1 µg of RNA was subjected to reverse transcription reaction with High-Capacity cDNA Reverse Transcription Kit (Thermo Fisher). Quantitative RT-PCR reactions were measured on a Viia 7 Real-Time PCR system (Thermo Fisher) using Power SYBR Green (Thermo Fisher) and transcript-specific primers below. Reactions were normalized to GAPDH levels for each biological replicate and relative transcript abundance was calculated with the comparative $C_t$ method.

| Gene Name | Forward Primer | Reverse Primer |
|---|---|---|
| ANKRD1 | GTGTAGCACCAGATCCATCG | CGGTGAGACTGAACCGCTAT |
| CTGF | TGGAGATTTTGGGAGTACGG | CAGGCTAGAGAAGCAGAGCC |
| CYR61 | CCCGTTTTGGTAGATTCTGG | GCTGGAATGCAACTTCGG |
| GAPDH | AATGAAGGGGTCATTGATGG | AAGGTGAAGGTCGGAGTCAA |
| AMOTL2 exon 7 | GGAATGTCGGATGAGAGTGG | GAAATCCAGCGGCTCTCTGAG |
| AMOTL2 exon 5 skipped | CTGTTCGTAGCTCTAAGATCCC | GAAATCCAGCGGCTCTCTGAG |
| AMOTL2 exon 5 included | GTAGGGAGAAAGAGGCTTG | GGTTTCTTCTTCCATGAAACAGGG |
| AMOTL2 exon 9 skipped | GCTACAGAGCTGTAGTCAG | GTCGGTGCCATCTGTTTTCG |
| AMOTL2 exon 9 included | CTGGAGACAGACAAGAACACAG | CAGAGCATTGTGCTTGGTTG |

## Semi-quantitative PCR

After 24 hr of compound treatment, cells were collected by trypsinization and subsequent centrifugation at 1200 × $g$. RNA was isolated using an RNeasy kit (Qiagen) and RNA concentrations were determined using a NanoDrop instrument. 1 µg of RNA was subjected to reverse transcription reaction with High-Capacity cDNA Reverse Transcription Kit (Thermo Fisher). 1 µL of cDNA was subjected to PCR with exon-specific primers using Platinum Taq DNA Polymerase (Thermo Fisher). PCR reactions underwent 35 cycles of replication. PCR reactions were visualized via horizontal gel electrophoresis using a 2% agarose gel. Gels were imaged using a ChemiDoc (BioRad, Hercules, CA).

| Gene Name | Forward Primer | Reverse Primer |
|---|---|---|
| AMOTL2 exon 5 | CTCCTGCTGCTGTTCGTAG | CTTCAACCGGGATCTTAGAG |
| AMOTL2 exon 9 | CTCTGGATGTAGCTACAGAG | CTGCTCGGCTGACTACAG |

## Plasmids

FLAG-tagged wild-type AMOTL2 (NM_016201) was obtained from OriGene, Rockville, MD. Codon-optimized sequences encoding truncated, FLAG-tagged transgenes of AMOTL2 and HA-tagged MAGI-1 WW domains were obtained from Integrated DNA Technologies (Coralville, IA) as gBlock HiFi Gene Fragments and cloned into the pCMV6 backbone via Gibson assembly using a HiFi DNA Assembly Cloning Kit (New England Biolabs, Ipswich, MA).

## Immunoblotting

HEK293A cells were plated in a standard growth medium at 500,000 cells per well in six-well plates and allowed to grow for 48 hr. Then, cells were treated with the indicated concentrations of SM04690 for 24 hr. Cells were washed with PBS and collected by the addition of 250 µL of RIPA buffer (EMD Millipore, Burlington, MA) with scraping. Lysates were clarified by centrifugation (18,253 × $g$, 5 min, 4 °C) and protein concentrations were determined by absorbance on a NanoDrop instrument (Thermo Fisher). Equal amounts of lysate were mixed with 4 x loading dye (2% SDS, 200 mM Tris, 20% glycerol, and 0.01% bromophenol blue) with ß-mercaptoethanol added to a final concentration of 10%. Protein material was resolved by SDS-polyacrylamide gel electrophoresis (SDS-PAGE) in 12-well gels (Invitrogen (Waltham, MA), 4–12% Bis-tris BOLT gels) in 0.9 x MOPS-SDS buffer (Invitrogen). Samples were

transferred to polyvinylidene difluoride (PVDF) membrane (Thermo Fisher) using a semi-dry transfer apparatus (BioRad). Membranes were blocked for 1 hr at room temperature with 5% non-fat dry milk (BioRad) in Tris-buffered saline (TBS, Corning) containing 0.1% Tween-20. Primary antibodies were incubated with shaking overnight at 4 °C in 5% milk in TBST or 5% bovine serum albumin (BSA) in TBST. Membranes were exposed to either fluorophore-conjugated secondary antibodies (Li-Cor (Lincoln, NE); 1:2000) or HRP-conjugated secondary antibodies (Sigma; 1:3000) in TBST with 5% milk for 30 min (Li-Cor) or 1 hr (HRP) at room temperature. Signals were recorded with either a Li-Cor fluorescence imager or exposed to HRP substrate (West Dura Substrate, Pierce (Waltham, MA)) and signals recorded using autoradiography film (Genesee Scientific, El Cajon, CA) or by imaging using a ChemiDoc instrument (BioRad).

| Antigen | Supplier | Catalog # | Dilution |
| --- | --- | --- | --- |
| Phospho-YAP (S127) | Cell Signaling Technologies | 13008 S | 1:1000 (BSA) |
| YAP (total) | Cell Signaling Technologies | 140074 S | 1:1000 (BSA) |
| AMOTL2 | Sigma | HPA063027 | 1:1000 (Milk) |
| HA | Cell Signaling Technologies | 3724 S | 1:1000 (BSA) |
| FLAG | Sigma | F1804 | 1:1000 (Milk) |
| Tubulin | Sigma | T6557 | 1:2000 (Milk) |

## Immunoprecipitation studies

Plasmids were transiently transfected to HEK293T cells for transgene expression using FuGENE (4 µL FuGENE per 1 µg of DNA) in 100 µL of Opti-MEM (Gibco) per well of a six-well plate (2 µg DNA total per well). Media was replaced 24 hr later. Following an additional 24 hr incubation, cells were washed with PBS and collected by the addition of 250 µL of RIPA buffer (EMD Millipore) or non-denaturing lysis buffer (20 mM Tris, 137 mM NaCl, and 0.5% NP-40) with scraping. For MAGI1 immunoprecipitation, lysates were tip sonicated. Insoluble material was separated by centrifugation (18,253 × $g$, 5 min, 4 °C) and the protein concentration in cellular lysates was determined by absorbance on a nanodrop instrument (Thermo Fisher). 1 mg of lysate in 1 mL of lysis buffer was incubated overnight at 4 °C with 20 µL of anti-FLAG M2 magnetic bead slurry (MilliporeSigma, no. M8823). After three washes with lysis buffer, immunoprecipitated material was eluted with 250 µg per mL of FLAG peptide (DYKDDDDK, Sino Biological (Beijing, China)). Eluted material from immunoprecipitations and whole-cell lysates were evaluated by immunoblotting analyses as described above.

## Imaging studies

For AMOTL2 localization studies, MDCK cells were plated in 12-well plates with glass coverslips at either 10,000 (sparse) or 400,000 (dense) cells in a growth medium and allowed to propagate for 48 hr. For treatment studies, 350,000 cells in growth medium propagated for 48 hr cells were treated for 24 hr with 1 µM SM04690. Growth medium was removed, and cells were fixed with ice-cold methanol for 10 min at –20 °C. After washing five times with PBS, cells were blocked and permeabilized with 5% FBS and 0.1% Triton-X in PBS for 30 min at room temperature. Primary antibodies in the same blocking buffer were incubated at 1:100 overnight at 4 °C. After three washes with PBS, Secondary AlexaFluor-conjugated antibodies (Life Technologies (Carlsbad, CA); 1:500 in 5% FBS, 0.1% Triton-X in PBS) with Hoechst 33342 dye were incubated for 1 hr at room temperature in the dark. The wells were washed three more times with PBS and then coverslips were removed and mounted onto glass slides using the ProLong Gold Antifade Mount (Thermo Fisher). Slides were imaged using a laser-scanning Zeiss (Oberkochen, Germany) confocal LSM 720 microscope and colocalization coefficients were calculated using Zeiss Zen software. For AMOTL2 overexpression localization studies, HEK293A cells plated in six-well plates with glass coverslips were transfected with 1 µg of AMOTL2 constructs and grown for 48 hr. Fixing, staining, and imaging proceeded with the same steps as above. Puncta were quantified using Image J (NIH, Bethesda, MD). For YAP localization studies, MDCK cells were plated in 12-well plates with glass coverslips at 250,000 cells per well and allowed to propagate for 48 hr. Cells were then treated for 24 hr with 1 µM SM04690. Growth medium was removed, and cells were fixed with ice-cold methanol for 10 min at –20 °C. After washing three times with PBS, cells were

permeabilized with 0.1% Triton X-100 for 15 min at room temperature. Primary antibodies in 5% BSA in PBS were incubated at 1:100 overnight at 4 °C. After three washes with PBS, Secondary AlexaFluor-conjugated antibodies (Life Technologies; 1:500 in 5% BSA in PBS) with Hoechst 33342 dye were incubated for 1 hr at room temperature in the dark. The wells were washed three more times with PBS and then coverslips were removed and mounted onto glass slides using the ProLong Gold Antifade Mount (Thermo Fisher). Slides were imaged using a laser-scanning Zeiss confocal microscope as above.

| Antigen | Supplier | Catalog # | Dilution |
| --- | --- | --- | --- |
| YAP (total) | Santa Cruz Biotechnology, Dallas, TX | SC-101199 | 1:100 |
| AMOTL2 | Sigma | HPA063027 | 1:100 |
| E-cadherin | Cell Signaling Technologies, Danvers, MA | 14472 | 1:100 |
| PALS1 | Santa Cruz Biotechnology | SC-365411 | 1:100 |
| MAGI-1 | Abnova, Taipei City, Taiwan | H00009223-M02 | 1:100 |

## Human keratinocyte proliferation assays

Immortalized human keratinocytes (HaCaT cells) were plated at $5 \times 10^3$ cells per well in 12-well plates in a reduced-serum-containing medium (DMEM containing 1% penicillin/streptomycin and 0 or 2% FBS). After 24 hr, cells were treated with the indicated concentrations of SM04690. After 7 days of growth under these conditions, cells were fixed in 4% PFA in PBS for 10 min, exposed to 2 µg per mL Hoechst 33342 for 10 min and imaged on a Nikon Eclipse Ti microscope. Nuclei were quantified by an automated ImageJ macro and reported as nuclei per imaging field.

## Chemicals

SM04690 was obtained from Selleck Chemicals, Houston, TX. T025 and CLK-in-T3 were from MedChemExpress, Monmouth Junction, NJ. CC671, Harmine, INDY, Leucettine L41, ML167, and TG003 were from Cayman Chemical, Ann Arbor, MI. All chemicals were dissolved in DMSO with no further purification.

## Statistics

All statistical analyses were univariate two-sided t-tests. All experiments were performed at least twice.

## Acknowledgements

This work was supported by the NIH (GM146865 to MJB). MLB was supported by a CIRM training fellowship (EDUC4-12811).

# Additional information

## Funding

| Funder | Grant reference number | Author |
| --- | --- | --- |
| California Institute for Regenerative Medicine | EDUC4-12811 | Maya L Bulos |
| National Institute of General Medical Sciences | GM146865 | Michael J Bollong |

The funders had no role in study design, data collection and interpretation, or the decision to submit the work for publication.

## Author contributions

Maya L Bulos, Data curation, Formal analysis, Validation, Investigation, Visualization, Methodology, Writing – original draft, Writing – review and editing; Edyta M Grzelak, Data curation, Formal analysis,

Investigation; Chloris Li-Ma, Data curation; Emily Chen, Data curation, Formal analysis; Mitchell Hull, Supervision, Methodology; Kristen A Johnson, Supervision, Project administration; Michael J Bollong, Conceptualization, Formal analysis, Supervision, Funding acquisition, Validation, Visualization, Methodology, Writing – original draft, Project administration, Writing – review and editing

**Author ORCIDs**
Maya L Bulos (ID) http://orcid.org/0009-0001-9002-4270
Michael J Bollong (ID) https://orcid.org/0000-0001-9439-1476

Reviewer #1 (Public Review): https://doi.org/10.7554/eLife.88508.3.sa1
Reviewer #2 (Public Review): https://doi.org/10.7554/eLife.88508.3.sa2
Reviewer #3 (Public Review): https://doi.org/10.7554/eLife.88508.3.sa3
Author Response https://doi.org/10.7554/eLife.88508.3.sa4

---

## Additional files

**Supplementary files**
• MDAR checklist

**Data availability**
All data generated or analysed during this study are included in the manuscript and source data files.

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
