## [Editor Report · eLife assessment]

This paper reports **important** findings on a potent activator of the YAP pathway, demonstrating its mechanism through alternative splicing changes. The authors provide **convincing** evidence to support their claims. This research is of interest to biologists studying alternative splicing or the Hippo pathway, with significant implications for medical research.

---

## [Referee Report · Reviewer #1 (Public Review)]

In order to find small molecules capable of enhancing regenerative repair, this study employed a high throughput YAP-activity screen method to query the ReFRAME library, identifying CLK2 inhibitor as one of the hits. Further studies showed that CLK2 inhibition leads to AMOTL2 exon skipping, rendering it unable to suppress YAP.

The novelty of the study is that it showed that inhibition of a kinase not previously associated with the HIPPO pathway can influence YAP activity through modification of mRNA splicing. The major arguments appear solid.

In the revised manuscript, additional discussion was provided regarding drug concentration and molecular mechanisms, which helps clear some of the confusing points in the original manuscript.

---

## [Referee Report · Reviewer #2 (Public Review)]

In this manuscript, the authors have screened the ReFRAME library and identified candidate small molecules that can activate YAP. They found that SM04690, an inhibitor of the WNT signaling pathway, could efficiently activate YAP through CLK2 kinase which has been shown to phosphorylate SR proteins to alter gene alternative splicing. They further demonstrated that SM04690 mediated alternative splicing of AMOTL2 and rendered it unlocalized on the membrane. Alternatively spliced AMOTL2 prevented YAP from anchoring to the cell membrane which results in decreased YAP phosphorylation and activated YAP. Previous findings showed that WNT signaling more or less activates YAP. The authors revealed that an inhibitor of WNT signaling could activate YAP. Thus, these findings are potentially interesting and important. However, the present manuscript provided a lot of indirect data and lacked key experiments.

Major points:

1. In Figure S3, since inhibition of CLK2 resulted in extensive changes in alternative splicing, why did the authors choose AMOTL2? How to exclude other factors such as EEF1A1 and HSPA5, do they affect YAP activation? Angiomotin-related AMOTL1 and AMOTL2 were identified as negative regulators of YAP and TAZ by preventing their nuclear translocation. It has been reported that high cell density promoted assembly of the Crumbs complex, which recruited AMOTL2 to tight junctions. Ubiquitination of AMOTL2 K347 and K408 served as a docking site for LATS2, which phosphorylated YAP to promote its cytoplasmic retention and degradation. How to determine that alternative splicing rather than ubiquitination of AMOTL2 affects YAP activity? Does AMOTL2 Δ5 affect the ubiquitination of AMOTL2? Does overexpression of AMOTL2 Δ5Δ9 cause YAP and puncta to co-localize?

2. The author proposed that AMOTL2 splicing isoform formed biomolecular condensates,.However, there was no relevant experimental data to support this conclusion. AMOTL2 is located not only on the cell membrane but also on the circulating endosome of the cell, and the puncta formed after AMOTL2 dissociation from the membrane is likely to be the localization of the circulating endosome. The author should co-stain AMOTL2 with markers of circulating endosomes, or conduct experiments to prove the liquidity of puncta to verify the phase separation of AMOTL2 splicing isoform.

3. The localization of YAP in cells is regulated by cell density, and YAP usually translocates to the nucleus at low cell density. In Figure 2E, the cell densities of DMSO and SM04690-treated groups are inconsistent. In Figure 4A, the magnification of t DMSO and SM04690-treated groups is inconsistent, and the SM04690-treated group seems to have a higher magnification.

4. There have been many reports that the WNT signaling pathway and the Hippo signaling pathway can crosstalk with each other. The authors should exclude the influence of the WNT signaling pathway by using SM04690.

---

## [Referee Report · Reviewer #3 (Public Review)]

This study on drug repurposing presents the identification of potent activators of the Hippo pathway. The authors successfully screen a drug library and identify two CLK kinase inhibitors as YAP activators, with SM04690 targeting specifically CLK2. They further investigate the molecular basis of SM04690-induced YAP activity and identify splicing events in AMOTL2 as strongly affected by CLK2 inhibition. Exon skipping within AMOTL2 decreases the interactions with membrane bound proteins and is sufficient to induce YAP target gene expression. Importantly, inhibitor concentrations that are sufficient to change YAP target gene expression show differential alternative splicing of AMOTL2. Overall the study is well designed, the conclusions are supported by sufficient data and represent an exciting connection between alternative splicing and the HIPPO pathway.

---

## [Author Response]

The following is the authors’ response to the original reviews.

**Public Reviews:**

**Reviewer #1 (Public Review):**
In order to find small molecules capable of enhancing regenerative repair, this study employed a high throughput YAP-activity screen method to query the ReFRAME library, identifying CLK2 inhibitor as one of the hits. Further studies showed that CLK2 inhibition leads to AMOTL2 exon skipping, rendering it unable to suppress YAP.The novelty of the study is that it showed that inhibition of a kinase not previously associated with the HIPPO pathway can influence YAP activity through modification of mRNA splicing. The major arguments appear solid.

We thank the Reviewer for their thoughtful assessment of this work. We have fully addressed each comment below in a point-by-point fashion.

There are several noteworthy points when assessing the results. In Figure S1C, 100nM drug was toxic to cells at 72 hours and 1nM drug suppressed cell proliferation by 60%. Yet such concentrations were used in Figure 1B and C to argue CLK2 inhibition liberates YAP activity (which one would assume will increase cellular proliferation). In Figure 1C it appears that 1nM drug treatment led to some kind of cellular stress, as cells are visibly enlarged. In Figure 1D, 1nM drug, which would have suppressed cell growth by 60%, did not affect YAP phosphorylation. Taken together, it appears even though CLK2 inhibitor (at high concentrations) liberates YAP activity, its toxicity may override the potential use of this drug as a YAP-activator to salve tissue regenerative repair, which was one of the goals hinted in the background section.

We do not claim that CLK2 inhibition is useful as a YAP activator, either as a precise pharmacological tool or as a therapeutic mechanism for inducing regenerative repair. Instead, the key finding of this work is to describe a novel, unanticipated cellular mechanism for activating YAP, one that should be considered when optimizing pharmacological candidates that modulate alternative splicing for diseases where potential proliferation is undesirable.

However, to address this point, we have included additional experimentation. Specifically, we show that cytotoxicity with compound treatment at 24 hours, a timepoint at which we perform most evaluation of alternative splicing induced by compound, is considerably less than that observed at 72 hours. Now included as Figure S1C, this panel shows while the compound displays some cytotoxicity at ~1 nM at 72 hours, the half maximal inhibitory potency at 24 hours is ~300 nM. As such, we believe there is not incongruity between YAP activity, cellular proliferation, and SM04690-induced cytotoxicity. It is simply such that higher concentrations of compound, and thus increased engagement of CLK2 and other targets of the inhibitor, result in a cumulative cytotoxic effect over time.

In Figure 2D, at 100nM concentration, the drug did not appear to affect AMOTL2 splicing. Even though at higher concentrations it did, this potentially put into question whether YAP activity liberated by this drug at 1nM (Fig 2A), 10-50nM (Fig 2C) concentrations is caused by altered AMOTL2 splicing. Discussions should be provided on the difference in drug concentrations in these experiments. Does the drug decay very fast, and is that why later studies required higher dose?

We believe this comment is in reference to Fig. 3D, and we argue that, while faint, there is the presence of AMOTL2 splicing at 100 nM SM04690 treatment as seen by a faint lower molecular weight band. However, to further understand the extent to which AMOTL2 is alternatively spliced in response to compound treatment, we performed RT-qPCR analysis of AMOTL2 splicing with an expanded concentration response. These results indicate that high magnitude exon skipping of AMOTL2 occurs starting at 10 nM with 24-hour treatment of compound (now in the manuscript as Fig. S4A). This result matches with our data in Fig. 2C, wherein YAP phosphorylation begins decreasing at 10 nM SM04690 treatment.

Likely impact of the work on the field: this study presented a high throughput screen method for YAP activators and showed that such an approach works. The hit compound found from ReFRAME library, a CLK2 inhibitor, may not be actually useful as a YAP activator, given its clear toxicity. Applying this screen method on other large compound libraries may help find a YAP activator that helps regenerative repair. The finding that CLK2 inhibition could alter AMOTL2 splicing to affect HIPPO pathway could bring a new angle to understanding the regulation of HIPPO pathway.
**Reviewer #2 (Public Review):**
In this manuscript, the authors have screened the ReFRAME library and identified candidate small molecules that can activate YAP. The found that SM04690, an inhibitor of the WNT signaling pathway, could efficiently activate YAP through CLK2 kinase which has been shown to phosphorylate SR proteins to alter gene alternative splicing. They further demonstrated that SM04690 mediated alternative splicing of AMOTL2 and rendered it unlocalized on the membrane. Alternatively spliced AMOTL2 prevented YAP from anchoring to the cell membrane which results in decreased YAP phosphorylation and activated YAP. Previous findings showed that WNT signaling more or less activates YAP. The authors revealed that an inhibitor of WNT signaling could activate YAP. Thus, these findings are potentially interesting and important. However, the present manuscript provided a lot of indirect data and lacked key experiments.

We thank the Reviewer for their thorough review of this work. We have responded to each comment below.

Major points:1. In Figure S3, since inhibition of CLK2 resulted in extensive changes in alternative splicing, why did the authors choose AMOTL2? How to exclude other factors such as EEF1A1 and HSPA5, do they affect YAP activation? Angiomotin-related AMOTL1 and AMOTL2 were identified as negative regulators of YAP and TAZ by preventing their nuclear translocation. It has been reported that high cell density promoted assembly of the Crumbs complex, which recruited AMOTL2 to tight junctions. Ubiquitination of AMOTL2 K347 and K408 served as a docking site for LATS2, which phosphorylated YAP to promote its cytoplasmic retention and degradation. How to determine that alternative splicing rather than ubiquitination of AMOTL2 affects YAP activity? Does AMOTL2 Δ5 affect the ubiquitination of AMOTL2? Does overexpression of AMOTL2 Δ5Δ9 cause YAP and puncta to co-localize?

AMOTL2 is the relevant cellular target, because among the entire transcriptome it was the third most alternatively spliced in response to CLK2 inhibition (Fig. S3). No other targets relevant to the Hippo pathway were identified.

We have shown that overexpression of exon skipped AMOTL2 (Fig. 3F) recapitulates the effect of compound, indicating that splicing per se is what drives the YAP activation phenotype. While AMOTL2 is ubiquitinated, these established sites of ubiquitination do not lie within exons 5 or 9. Thus, we anticipate that ubiquitination is less likely a driving factor in the observed phenotype. The manuscript is written as not to exclude this as a possibility, but it is downstream of what we describe, and we believe out of scope to explore this further in this preliminary report.

1. The author proposed that AMOTL2 splicing isoform formed biomolecular condensates. However, there was no relevant experimental data to support this conclusion. AMOTL2 is located not only on the cell membrane but also on the circulating endosome of the cell, and the puncta formed after AMOTL2 dissociation from the membrane is likely to be the localization of the circulating endosome. The author should co-stain AMOTL2 with markers of circulating endosomes or conduct experiments to prove the liquidity of puncta to verify the phase separation of AMOTL2 splicing isoform.

We do not claim AMOTL2 forms biomolecular condensates. Instead, we hypothesize in the Discussion section that AMOTL2 could possibly phase separate into biomolecular condensates based on its similarity to AMOT, which has been shown to phase separate and form cytoplasmic puncta (PMID: 36318920). AMOT has also been shown to colocalize with endosomes (PMID: 25995376), which also appear as puncta.

1. The localization of YAP in cells is regulated by cell density, and YAP usually translocates to the nucleus at low cell density. In Figure 2E, the cell densities of DMSO and SM04690-treated groups are inconsistent. In Figure 4A, the magnification of t DMSO and SM04690-treated groups is inconsistent, and the SM04690treated group seems to have a higher magnification.

In immunofluorescence experiments, cells were plated at the same density and grown for the same amount of time before treatment. Additionally, within an experiment, images were taken at the same magnification. Any apparent differences in cell density are due to effects of the compound.

1. There have been many reports that the WNT signaling pathway and the Hippo signaling pathway can crosstalk with each other. The authors should exclude the influence of the WNT signaling pathway by using SM04690.

While the WNT pathway has been shown to influence Hippo pathway activity, we have shown a direct effect of CLK2 inhibition by SM04690. Any WNT potential pathway effects are in addition to the splicing-based mechanism we described.

**Reviewer #3 (Public Review):**
This study on drug repurposing presents the identification of potent activators of the Hippo pathway. The authors successfully screen a drug library and identify two CLK kinase inhibitors as YAP activators, with SM04690 targeting specifically CLK2. They further investigate the molecular basis of SM04690-induced YAP activity and identify splicing events in AMOTL2 as strongly affected by CLK2 inhibition. Exon skipping within AMOTL2 decreases the interactions with membrane bound proteins and is sufficient to induce YAP target gene expression. Overall the study is well designed, the conclusions are supported by sufficient data and represent an exciting connection between alternative splicing and the HIPPO pathway. The specificity of the inhibitor towards CLK2 and the mode of action via AMOTL2 could be supported by further data:

We thank the Reviewer for their close examination of our work. We respond below.

1. The inconsistent inhibitor concentrations and varying results reported in the paper can be distracting. For instance, the response of endogenous targets to 100 nM concentration is described as a >5-fold increase in Figure 2B, whereas it is reported as a 1-1.5-fold response to 1000 nM in Figure 2D. This inconsistency should be addressed and clarified to provide a more accurate and reliable representation of the findings.

In Figure 2D, we have transduced cells with lentivirus, which most likely suppresses their responsiveness to compound treatment. We have addressed the issue of varying inhibitor concentrations in response to Reviewer 1.

1. In the absence of a strong inhibitor induced YAP target gene expression (Figure 2D), it is difficult to conclude the dependency on YAP expression, as investigated by siRNA mediated knockdown. In a similar experiment, the dependency of the inhibitor on CLK2 expression could be confirmed

While the sample with Scramble virus does not respond to the same extent that WT HEK293A cells do (e.g., Fig. 2B), there is still responsiveness to compound. Likewise, YAP knockdown cells display statistically significant decreases in YAP-controlled transcripts. This decrease of transcript is therefore sufficient evidence that SM04690 requires YAP for its activity. We have shown that multiple CLK2 inhibitors recapitulate the effect of SM04690, abrogating the need to show dependency of CLK2.

1. To further support the conclusion that CLK2 is the direct target of SM04690, it would be informative to investigate the effects of CLK1/4 inhibition on AMOTL2 exons (for example within RNA-seq data). If CLK1/4 inhibitors do not induce changes in AMOTL2 exons, it would strengthen the evidence for CLK2's role as the direct target. Including the results in the discussion would enhance the comprehensiveness of the study.

We showed that CLK1/4 inhibition with small molecules ML167 and TG003 does not affect YAP activity in our luciferase reporter assay (Fig. S2D), which we believe is sufficient evidence that CLK1/4 is neither the direct target of SM04690 nor relevant to the splicing mechanism we describe.

1. It would be important to determine the specific dose of SM04690 required to induce changes in AMOTL2 splicing. The authors observe that AMOTL2 protein levels appear unaffected at doses below 50 nM in Figure 3D, while YAP target genes are already affected at 20 nM in Figure 3G. Although Western blotting may not be the most sensitive method to detect minor changes in splicing, performing PCR experiments at lower doses could provide more insight into the splicing changes. Therefore, it is suggested that the authors include PCR experiments at lower doses to determine if changes in splicing are visible and to better establish the relationship between splicing and gene expression changes.

We agree with the Reviewer that this experiment is essential to better understand splicing changes with SM04690 treatment. Accordingly, we have added RT-qPCR-based analysis of AMOTL2 exon inclusion at lower concentrations between 10 nM and 100 nM (Fig. S4A). We included a similar discussion in response to a point from Reviewer 1.

**Reviewer #1 (Recommendations For The Authors):**
As stated in the public review section, it will be helpful to discuss the differences in drug concentration. Although no one should require or expect a perfect drug dose match throughout any study, in this study the drug dose clearly demarcated when CLK2 inhibitor help/hurt proliferation, when CLK2 inhibitor was able to affect YAP phosphorylation, and when CLK2 inhibitor was able to affect AMOTL2 splicing. This is not to challenge the major conclusions of the paper, but it is hard to ignore if no discussion is provided.Several suggestions on data presentation:1. Scale bar information is missing in Fig. 2E, 4A and 4B.

We have corrected this mistake in the revised manuscript.

1. For Fig.3 D and 3E, it's better if kD information was labeled alongside the AMOTL2 Western blot.

Thank you for the suggestion; we have added the appropriate labeling.

1. It's better to label Figure2D as sh YAP-1, sh YAP-2; Figure 3A as sh CLK2-1, sh CLK2-2 etc. Currently they are all labeled shRNA-1, shRNA-2, which can be confusing.

We have altered the labeling for clarity as requested.

**Reviewer #3 (Recommendations For The Authors):**
1. The use of asterisks in Figure 2D is unclear, especially their placement on the "Scramble" sample.

We have amended the asterisks and have also added more detail to the figure legend.

1. When designing primers for splicing-sensitive PCR, it is recommended that the skipping isoform is larger than 100 bp. This will help to avoid quantitative issues with ethidium bromide staining. In the results part, the text reads as if only the skipping isoform is present after SM04690 treatment.

This experiment was performed to confirm the presence of exon skipping in the treated samples. Accordingly, we did not optimize the ethidium bromide staining of the lower bp bands. We will take the size of the isoform into consideration in any future experiments. We thank the reviewer for catching the textual error and have amended the text in the manuscript.